# The Cytotoxicity Effect of Resveratrol: Cell Cycle Arrest and Induced Apoptosis of Breast Cancer 4T1 Cells

**DOI:** 10.3390/toxins11120731

**Published:** 2019-12-13

**Authors:** Hong Wu, Liang Chen, Feifei Zhu, Xu Han, Lindan Sun, Keping Chen

**Affiliations:** Institute of Life Sciences, Jiangsu University, 301 Xuefu Road, Zhenjiang 212013, China; hongwu@stmail.ujs.edu.cn (H.W.); oochen@ujs.edu.cn (L.C.); feifzhu@ujs.edu.cn (F.Z.); xuhan@stmail.ujs.edu.cn (X.H.); sunlindan@ujs.edu.cn (L.S.)

**Keywords:** resveratrol, cytotoxicity, RNA-seq, cell cycle, apoptosis

## Abstract

Resveratrol, a natural polyterpenoid, can scavenge reactive oxygen species in vivo to carry out the functions of antioxidation and antiaging. Resveratrol’s anti-cancer capability has attracted widespread attention, but its molecular mechanism has not been systematically explained. In this study, by comparing the activity of normal cell lines and cancer cell lines after treating with resveratrol, it was found that resveratrol has more significant cytotoxicity in cancer cell lines. Resveratrol could play a toxic role through inducing apoptosis of the cancer cell in a time- and concentration-dependent manner. A total of 330 significantly differential genes were identified through large-scale transcriptome sequencing, among which 103 genes were upregulated and 227 genes were downregulated. Transcriptome and qRT-PCR data proved that a large number of genes related to cell cycle were differentially expressed after the treatment of resveratrol. The changes of cell cycle phases at different time points after treating with resveratrol were further detected, and it was found that the cells were arrested in the S phase because of the percentage of cells in S phase increased and cells in G1/G0 phase decreased. In conclusion, resveratrol can inhibit the proliferation of 4T1 cancer cells by inhibiting cell cycle and inducing apoptosis.

## 1. Introduction

Second only to cardiovascular disease, cancer is the most serious threat to human life and health [1]. In recent years, significant progress in the monoclonal antibody field and the controversial CAR-T cell therapy has made the traditional small molecule drug seemingly less competitive. However, considering the economic benefits, malignant metastasis, and recurrence frequency, as well as other clinical factors, traditional drug therapy is still necessary [2]. Due to its low toxicity and side effects, drugs from natural plants are often used as synergistic drugs in cancer therapy [3]. Resveratrol (RSV), a polyphenol phytoalexin, was first isolated from the root of resveratrol [4,5]. Resveratrol from various sources, including grapes, blueberries, cranberries, mulberries, and peanuts, has been studied as a potential therapeutic drug [6,7]. A large number of experiments indicate that resveratrol has strong anti-inflammatory, antioxidation, and anti-cancer activities [8]. Therefore, its role against various diseases has received widespread attention, including cardiovascular, diabetes, asthma, kidney, liver, and cancer [9]. The high anti-cancer activity of resveratrol and its derivatives was reviewed systematically [10,11]. The results of antibacterial experiments showed that resveratrol had a strong inhibitory effect on bacteria such as *Staphylococcus aureus*, *Escherichia coli*, *Pseudomonas aeruginosa* and fungi [12]. On the other hand, resveratrol could reduce platelet adhesion and alter platelet activity during the anti-inflammatory process [13]. Resveratrol is also reported to promote metabolism and reduce oxidative stress, which can also be used as an antioxidant affecting the synthesis of nitric oxide that regulates DNA damage, cell cycle, apoptosis, and proliferation [14]. In addition, a number of studies on resveratrol have revealed possible mechanisms of UV protection, such as inhibiting the activation of NF-κB and preventing the expression of MMP-9 [15,16]. Resveratrol has gradually been found to have potential health benefits, including antiaging, anti-diabetes, anti-cancer, and anti-dementia [17,18]. Most of these studies are limited to animal models, and the relevant verification in humans is still in the early stage. Therefore, scientists analyzed the levels of resveratrol ingested and the overall mortality of various chronic diseases in 2014. It turned out that dietary intake of resveratrol was not significantly associated with longevity, inflammation, cancer, or cardiovascular health, which faded the legendary benefits of resveratrol. Recently, scientists have discovered that caraphenol A, a trimer of resveratrol, plays a unique role in gene therapy, which has brought resveratrol back into the spotlight. Torbett et al. found that caraphenol A safely enhanced the gene delivery efficiency from LVs (lentiviral vector) to the HSC (hematopoietic stem cell), also reducing the transmembrane protein-mediated restriction to making it easier for vectors to pass through, which is a possible way to improve the therapeutic effect of gene therapy [19].

In order to get a more comprehensive and mechanistic understanding of the toxic effect of resveratrol on cancer cells, the viability and apoptosis of cancer cells were detected from cellular and molecular levels. This study calculated IC50 (50% inhibiting concentration) of resveratrol in 4T1 breast cancer cell lines by detecting cell metabolic activity. It was demonstrated that resveratrol can induce apoptotic cell death. Transcriptome profiles of the breast cancer cells were used to screen genes closely associated with RSV treatment. Through analyzing the differentially expressed genes between treated and control groups, which were functionally annotated and pathway enriched, it was found that the differentially expressed genes were tightly associated with apoptosis and cell cycle. Finally, different cycle phases were detected to explain the possible molecular mechanism of RSV in inhibiting proliferation and inducing apoptosis of the 4T1 cells.

## 2. Results

### 2.1. Resveratrol Significantly Inhibits the Proliferation of Cancer Cells

We evaluated the cytotoxic effect of resveratrol on two types of normal cell lines (the renal tubular epithelial cell line HK-2 and normal human liver cell line L02), and two types of tumor cell lines (hepatocellular carcinoma HepG2 and murine mammary carcinoma cell line 4T1) (Figure 1). For normal cells, low concentration (50 µM) and short time (24 h) treatment had no significant effect on cell viability with a slight increase in L02 cells (Figure 1A). After treating for longer, a significant inhibitory effect, which is dose-dependent, appeared (Figure 1B,C). For cancer cells, resveratrol can significantly decrease cell viability in a dose-dependent manner all the time. By comparison, it was found that resveratrol had a more obvious toxic effect on cancer cells compared to normal cells, especially on 4T1 cells. Therefore, the 4T1 cell line was chosen for further study.

In order to evaluate the effect of resveratrol on 4T1 cells, the time-dependent proliferation curve (Figure 2A) and dose-dependent viability curve (Figure 2C) were drawn. As can be seen from the proliferation curve, the proliferation rate decreased with the increase of treatment dose. When the treatment concentration was higher than 100 µM, 4T1 cells showed inhibitory growth (Figure 2A). The viability of 4T1 cells also decreased with the increase of dose and time. After being treated for 72 h, the viability decreased significantly, and only 20% were viable at over 150 µM (Figure 2B), indicating that the cells have lost their ability to proliferate at these concentrations. According to the dose-dependent viability curve, the IC50 at 72 h was calculated to be 93 µM (Figure 2C).

### 2.2. Effects of Resveratrol on the Cell Morphology

After the treatment of resveratrol, the morphological changes of 4T1 at different time periods were observed with inverted fluorescent microscope. With the increase of treatment concentration, the cell becomes swollen with blurred edges, disrupted membranes, and decreased density, and there are fewer contact areas between cells. Under the low treatment dose (<50 µM), proliferation and growth situation was basically not affected, and cell adhesion was appropriate along with the treatment time; however, when treated with high concentration (>100 µM), obvious inhibition of adherent growth can be observed, and cells were unable to proliferate, accelerating the death of 4T1 cells (Figure 3A). Furthermore, with the increase of treatment time, the area of most observed cells gradually decreased at each concentration, except for several swollen and ruptured cells (Figure 3B).

### 2.3. Resveratrol Induced Apoptosis of 4T1 Cells

#### 2.3.1. Detection of Cell Apoptosis

Acridine orange (AO) is a vital dye and will stain the nuclei of both live and dead cell to green while ethidium bromide (EB) will stain only cells that have lost membrane integrity to red. Thus, live cells will appear uniformly green while early apoptotic cells will have condensed or fragmented nuclei with bright green color. Late apoptotic cells will show condensed and fragmented orange chromatin. The results showed that increased resveratrol concentrations resulted in gradual increases in orange and red staining accompanied by reductions in green staining of nuclei (Figure 4A), indicating cell damage and apoptosis. By counting the number of red, orange, and green cells, the apoptotic rate was calculated (Figure 4B). When treated with 50 µM resveratrol for 24 or 48 h, the apoptotic rate remained between 1% and 6%, which has no significant difference compared with control group (0 M), while treatment for 72 h, significant difference occurred with apoptotic rate of 20%; furthermore, with 100 and 150 µM resveratrol, cell apoptosis rate increased to 40% and 70%, respectively; finally, under 200 and 250 µM resveratrol treatment, apoptotic rate increased from 75% to 90%. Therefore, high concentrations (>150 µM) of RSV could cause serious membrane damage in around 85% of cells. These results indicate that apoptotic rate gradually increase with the RSV concentration and treatment time. It is verified that at around 100 µM RSV can induce half of the cells to undergo apoptosis at 72 h, which is consistent with the previous IC50 results detected by CCK-8. 

#### 2.3.2. Concentration Dependent Apoptosis Assay

As necrotic cells are also orange in AO/EB staining, to differentiate apoptotic cells from necrotic cells, the cells treated with different concentrations of resveratrol (50, 100, 150, 200, and 250 µM) for 48 h were also stained with Annexin V and PI, and then distinguished by flow cytometry. The results in Figure 5 show that with the increase of treatment dose, the proportion of normal cells decreased gradually along with the increase of late apoptotic cells, and mechanical necrotic cells remained below 5%. By calculating the overall apoptotic rate, it was found that about 25% of the cells were apoptotic with the treatment of 50 µM resveratrol for 48 h, which was significantly different from the 6% of the control group. It could be observed that 100 µM resveratrol brought about 75% apoptotic rate, and higher concentrations after that caused almost all cells to be damaged. Compared with the results of flow cytometry detection and morphological detection, it was found that under the same situation, treatment of 100 µM resveratrol for 48 h, showed 50% apoptotic rate, while the apoptotic rate detected by flow cytometry was about 80%, which appears to contradict the result of calculated IC50. There was a great difference in apoptotic rate detected by two methods. Therefore, in order to determine the specific situation, IC50 (93 µM) was used in the following experiments to determine the apoptosis stages at different time points.

#### 2.3.3. Time Dependent Apoptosis Assay

After being treated with IC50 (93 µM) for different times (0, 12, 24, 36, and 48 h), 4T1 cells were stained by AO/EB mixture solution. The results of fluorescence observation showed that more and more cells were stained red as the processing time increased (Figure 6A). The percentage of apoptotic cells was calculated according to the stained picture (Figure 6C), and it was found that the treatment of IC50 for 48 h could cause morphological apoptosis of about 30% cells. In addition, flow cytometry was also used to detect Annexin V/PI stained cells after treatment for different times (0, 12, 24, 36, and 48 h). The four-quadrant diagram was drawn to distinguish mechanical necrotic cells, normal cells, early apoptotic cells, and late apoptotic cells. The results showed that, compared with the control group, the ratio of normal cells decreased gradually, and the proportion of early and late apoptotic cells increased gradually with the increase of treatment time (Figure 6B). Furthermore, the proportion of normal cells and apoptotic cells in the control group and the IC50 treatment group was calculated (Figure 6D), it was found that the two types of cells in the control group and the IC50 group showed significant differences from 24 h. When the treatment time reached 48 h, about 50% of 4T1 cells were in apoptotic state. In comparison, it was found that after 48 h of IC50 treatment, AO/EB staining indicated that apoptosis rate was about 30% of cells, while Annexin V/PI staining combined with flow cytometry detection found that the apoptosis rate was about 50%, with a fairly large difference between these two methods. The difference may be generated by different cell states and operational deviations. However, no matter which method was used, it could be confirmed that the IC50 of resveratrol inhibited the proliferation of 4T1 cells and induced apoptosis.

### 2.4. Transcriptome Analysis of 4T1 Cells 

As demonstrated above, 4T1 cells showed morphological changes in a dose- and time-dependent manner when treated with RSV, showing decreased activity, proliferation rate, and increased apoptosis. In order to determine the molecular mechanism of these changes, RNA sequencing was performed on 4T1 cells with IC50 culture (RSV_treated group) and normal culture (Mock group) at 24 h. All samples were measured independently in triplicate. The differentially expressed genes were analyzed to infer the candidate genes related to cell proliferation and apoptosis affected by RSV treatment, and to reveal the possible function of these differential genes and related molecular mechanisms.

#### 2.4.1. Screening and Functional Enrichment Analysis of Differential Genes

The expression level of genes and transcripts were quantitatively analyzed to facilitate the subsequent analysis of the differential expression of genes between different samples, and the regulatory mechanism of genes could be revealed by combining sequence and functional information. In each sample, the logarithmic value of FPKM is between 1.16 and 1.18 (Figure 7A), indicating that the expression of these genes is in a higher level. The Venn diagram showed that the percentage of co-expressed genes were 93.6% between the two groups, while 508 and 354 genes were specifically expressed in the Mock group and RSV_treated group, respectively (Figure 7B). The correlation coefficients between the six samples were all above 0.97 (Figure 7C), indicating high similarity of gene expression, which proved that the variation between biological replications was very low. Principal component analysis (PCA) showed that that Mock3 and RSV_t1 are outlier samples, so subsequent analysis of expression difference excluded these two samples (Figure 7D).

Differential expression analysis was performed on the remaining four samples to identify the differentially expressed genes (DEGs) between groups, so as to further study the function of these genes. A total of 330 genes with significantly different expressions were identified, among which 103 genes were upregulated and 227 genes were downregulated compared to the control group. Based on whether the gene is present in a certain pathway, it is possible to understand the biological process in which the DEGs may be involved, and can assist to understand the relationship between the genes and RSV induced pathway changes. In order to study the regulatory mechanism of DEGs and their status in biological metabolism, the gene annotation results were filtered and screened through functional enrichment to obtain more meaningful functional information. Finally, the GO and KEGG enrichment string diagrams were made to show the top-10 entries of GO Term or KEGG Pathway (Figure 8). The first 10 entries of GO Term are regulation of cell cycle, regulation of cell division, regulation of cell cycle process, cellular response to organic substance, cellular response to chemical stimulus, mitotic cell cycle process, cell division, cell cycle, regulation of mitotic nuclear division, and regulation of nuclear division; the top-10 KEGG Pathways are AGE-RAGE signaling pathway in diabetic complications, PI3K-Akt signaling pathway, cell cycle, focal adhesion, relaxin signaling pathway, amphetamine addiction, Ras signaling pathway, FoxO signaling pathway, and fluorobenzoate degradation. It can be found that the key GO terms are cell cycle, cell division, and cell adhesion, which indicates that the treatment of RSV has a significant impact on the cell cycle of 4T1 cells. Therefore, we continued to verify and explore whether resveratrol affects cell cycle and related gene expression later.

#### 2.4.2. qRT-PCR Validation of Differentially Expressed Genes

Based on the cluster analysis results of GO and KEGG, genes related to apoptosis, cell cycle and cell division were selected for qRT-PCR to verify the trend of gene expression level. The following principles should be observed when selecting genes: differential expression multiples (log2FC) should be large between samples (1.2 ≤ log2FC ≤ 1.5); genes should be highly expressed, that is, the value of FPKM should at least more than 50 in each sample; the depth of sequencing (read count) should be relatively high. Based on the above three filters, the following genes were selected: *Gadd45b, Cdc6, Dusp9, Amhr2, Cirbp, Hnrnpal, Trmt61a, Hmga2, J02Rik, Col6a1, Cdc20, Lama5, Fos, Pttg1, Vegfa, Dusp1, Cdc25b, Aurka, Plk1, Ccnb2, Kif23, Bub1b, Pck2, Ccnb1, Atf4, Col6a3*, and *Esp11*. The results showed that the expression trend of differential genes verified by qRT-PCR was mostly consistent with the RNA-seq results (Figure 9), which indicated that the transcription abundance of these DEGs in signal transduction was highly reliable. However, there are also some gene expression trends that are inconsistent (green oblique line), which are probably because RNA-seq is a large-scale screening tool to reflect the overall gene expression trend of the samples, and it cannot be guaranteed that the expression trend of every gene is consistent with qRT-PCR. In other words, RNA-seq and qRT-PCR themselves are two different experimental platforms, and the results may not necessarily agree with each other completely.

### 2.5. Interaction Network Analysis of Resveratrol’s Protein Targets

The transcriptome analysis showed that resveratrol may inhibit cell division and induce apoptosis by arresting the cell cycle of 4T1 breast cancer cells. In order to further analyze the possible mechanisms or pathways, related protein targets of resveratrol were identified in Traditional Chinese Medicine Systems Pharmacology Database and Analysis Platform (TCMSP) [20], and then these targets were uploaded to online STRING software [21], so as to obtain the interaction information between target proteins. Finally, the analysis data was imported into Cytoscape3.2.1 software to construct the protein-protein interaction network (PPI network, Figure 10). In this interaction network, the Degree of each node represents the number of lines connected to this target. Therefore, the larger the Degree is, the stronger the interaction relationship between the targets is, and it also indicates that this key target protein plays a pivotal role in the whole interaction network. As can be seen from the bottom of Figure 10, cell cycle related genes like Cdc, Cdk, Ccnd and apoptosis related genes like Brca, Bcl, Bax are obviously the key targets. Therefore, the PPI results verified the transcriptional results that the cell cycle related genes played a crucial role in the effect of resveratrol, which was obviously in line with our expectation of finding downstream targets related to apoptosis. The distribution of the cell cycle changed by resveratrol will be discussed in the following experiments.

### 2.6. Resveratrol Changed the Phase Distribution of the Cell Cycle

The prominent features of apoptosis are DNA fragmentation and damage. In order to determine whether the apoptotic cell induced by RSV has altered cell cycle distribution, we further analyzed the cell cycle under RSV treatment by flow cytometry. The fluorescent dye PI could enter cells having compromised membranes, and bind to DNA stoichiometrically, allowing evaluation of cell proportions in each phase based on their DNA levels. By comparing the cell proportions at each phase at different time points (Figure 11), it can be seen that the percentage of cells in G1/G0 phase gradually decreased from 60% to 38%, and that in S phase gradually increased from 13% to 25%, while the cell proportion in G2/M phase significantly decreased. Moreover, when analyzing the difference between these data, it was found that the percentage of cells in different phases showed significant differences starting at 24 h. Therefore, resveratrol can promote the transformation from G1 to S phase in breast cancer cells, and then induce cycle arrest in S phase, so that their proliferation capacity weakened, and cell viability reduced.

## 3. Discussion

Previous studies have shown that resveratrol can inhibit the proliferation of breast cancer cells [22], and studies in vivo have also demonstrated that it can inhibit the growth of tumors [23,24]. When resveratrol was used in combination with other classic drugs, cancer cells would be killed more dramatically [25]. The treatment effect depends on the RSV concentrations. Specifically, resveratrol of low dosage could stimulate the growth of ERα+ (estrogen receptor α positive) breast cancer cells, while it always acts as an inhibitor in ERα- (estrogen receptor α negative) breast cancer cells by acting as an estrogen-metabolizing inhibitor to affect the proliferation of different types of breast cancer cells [26]. Bove et al. found that 30 µM RSV could completely inhibit the proliferation of 4T1 cells [27]. Similarly, Lee et al. set the concentration gradient in the range of 0–30 µM to study the effect of RSV on 4T1 cells [28]. However, the calculated IC50 is 93 µM in this study, which is quite different from previous studies. In our study, mitochondrial enzyme activity was quantitatively measured via cell viability assay, which was more accurate than direct visual observation of counting cells. Of course, different cell states also have a great influence on the results. Therefore, practical considerations for proper RSV concentrations for a particular cell experiment are needed. Only in this way can it be potentially applied to the clinical treatment of various diseases. On the issue of RSV induced apoptosis, one study has shown that RSV could induce phosphorylation of p53 through the activation of mitogen-activated protein kinase (ERK1/2) and nuclear translocation [29]. In another study, it was indicated that the analogues of resveratrol could influence cell cycle status and cell cycle distribution, thereby inducing apoptosis [30]. Unfortunately, these experiments did not provide evidence of changes related to cell cycle at molecular level.

When detecting cell proliferation at the cellular level, CCK8 staining and Brdu (the thymidine analogues) staining are usually used. However, these end-point assays need to destroy cells, leading to cell death or the destruction of cell structure. The real time cell analyzer (RTCA, xCELLigence, Roche) is an impedance-based technology that can be used for label-free and real-time monitoring of cell properties, such as cell adherence, proliferation, migration, and cytotoxicity, which could have widespread use [31]. In the detection of apoptosis at the molecular level, the normally detected proteins are caspase-9, caspase-3, and PARP (substrates of caspase-3), located in the mitochondrial pathway [32]. In addition, the ratio of Bcl-2/Bax could also be used to represent the degree of apoptosis [33]. In terms of monitoring the cell cycle, in addition to the flow cytometry analysis of PI staining used in this study, immunostaining evaluation of cell cycle markers is also an effective method, among which MCM2, ki-67, and PCNA are commonly used [34]. Compared with PCNA and ki-67, the advantage of MCM is that it is expressed throughout the cell cycle and it is immune to DNA damage repair and other external factors [35]. Therefore, MCM protein is considered to be an ideal marker of cell proliferation superior to PCNA and ki-67. However, from the practical application level, PCNA and ki-67 are more widely used. Unfortunately, the methods used to detect cell proliferation, apoptosis, and cell cycle in this paper are relatively simple. In future experiments, different methods mentioned above should be used in an attempt to achieve more profound conclusions.

In this study, it was found that cells in good growth conditions and adequate nutrition were generally in a spindle shape, and the sharp feeler was probably conducive to the adherent growth of cells. On the other hand, cells treated with resveratrol would become round with a blurred cell membrane boundary, diminished sharp feeler formation, and even become floating (Figure 3). The results showed that cell adhesion changed significantly after RSV treatment. After the treatment of RSV, apoptosis of 4T1 cells at cellular level was verified (Figure 5 and Figure 6). At molecular level, transcriptome analysis showed that a large number of significantly DEGs were indeed enriched in apoptosis-related pathways as well as in cell cycle pathways. The highlighted genes/proteins in red in Figure 12 are the DEGs with significant changes in cell cycle pathways. Actually, cell adhesion and morphological changes are closely related to the cell cycle progression. Before cell division, cells have to disassemble adhesion complexes, evacuate from extracellular matrix (ECM), and round up [36,37,38,39]. Rounding up, a major shape change that happened in M phase [40,41] to provide enough space for the accurate formation of the spindle and the condensation of chromosomes [42,43,44], could be regulated through the decomposition of adhesion complexes [41]. In addition, adhesion complexes contain phosphorylation sites of mitotic kinase [45], so these cycle-related enzymes may be directly involved in regulating cell adhesion. All the results discussed above directly proved the tight connection among cell adhesion complex, morphological changes of cells, and cell cycle progression, while Jones et al. further demonstrated that CDK1 was the bridge that maintained the relationship among the three [46]. Specifically, the complex of CDK1 and cyclin A2 promotes the formation of adhesion complexes and cytoskeleton in interphase. When cells transit from G1 phase to S phase, the area of adhesion complexes increased in a CDK1-dependent manner. However, in G2 phase, the increased levels of cyclin B1 and the inhibition of CDK1 by Wee1 both caused a reduction in the area of the adhesion complex.

Weakened intercellular adhesion could lead to a significant increase in the movement ability of cancer cells, which cannot be achieved without the strong support of the cytoskeleton. The main components of cytoskeleton are microtubules, microfilaments, and intermediate fibers, while microtubulin is the main component of spindle filaments in mitotic phase. The results of transcriptome analysis in our research revealed that genes related to the synthesis of cohesin like Cdc20, Pttg, Esp1, and BubR1 are differentially expressed in M phase (Figure 12). Cohesin would contribute to the combination of spindle and chromosome in mitosis. Therefore, it could be speculated that alterations in the synthetic pathway of cohesin may lead to changes in cell adhesion. 

Cyclins could activate different cyclin dependent kinases (CDKs) to accelerate the cycle progression, while cyclin dependent kinase inhibitor (CKI) acts as a brake to inhibit the activity of CDK and Cyclin-CDK complex. As shown in Figure 12, p19, differentially expressed in G1 phase, is a kind of CKI belonging to the Ink4 family, which can inhibit the activity of Cyclin D-CDK4/6 complex. The regulation of extracellular signal includes stimulation signal and inhibition signal. On the other hand, the regulation of extracellular signals includes stimulation and inhibition. In G1 phase, growth stimulating factors transmit proliferation signals into the cells through activating GSK3β and phosphorylate Rb through the further activation of CDK4/6, which separates Rb from transcription factor E2F to promote the combination of Cyclin E and CDK2, so as to enter the S phase for DNA synthesis [47]. Whereas, TGFβ as a kind inhibitory signal can downregulate the expression of CDK and Cyclin by facilitating the generation of CKI to arrest cells in G1 phase [48]. Similarly, the transcription factor P53 blocks cells in G1 phase by a similar pathway. In addition, P53 could also mediate DNA repair and apoptosis through Gadd45 and Bax, respectively. In general, the signaling pathways mediated by P53 are a complex biological progression in G1 phase. Previously mentioned CDK1 mainly affects cell cycle progression by binding to cyclins A2 and B1 [49]. In G2 phase, because CDK1 is phosphorylated by Wee and related kinases, the activity of cyclin B1-CDK1 complex is inhibited, so as to prevent premature entry into M phase [50,51]. Therefore, the main function of Cyclin B1 is to regulate the entry of cells into the M phase [52,53,54]. Once the cells enter into prophase of division, the activity of Cyclin B1-CDK1 will gradually increase [55], and then the activated Cyclin B1-CDK1 will be transferred into the nucleus [56] to induce cell division.

## 4. Conclusions

In summary, the IC50 of resveratrol in 4T1 cells was calculated to be 93 µM. Resveratrol could induce apoptotic cell death in a dose- and time-dependent manner. Transcriptome profiles was used to screen genes closely associated with RSV treatment, which found that DEGs were functionally annotated and pathways enriched in the terms of apoptosis and cell cycle. The 4T1 cells were arrested in the S phase according to the cell cycle distribution.

## 5. Materials and Methods 

### 5.1. Preparation of Resveratrol

Resveratrol (Sigma-Aldrich, Shanghai, China) was dissolved in dimethyl sulfoxide (DMSO) at 100 mM and stored at −20 °C. The concentrations used in this study were 0, 50, 100, 150, 200, and 250 µM, freshly diluted in RPMI-1640 or DMEM medium before use. Controls were always treated with the same amount of DMSO as used in the corresponding experiments.

### 5.2. Cell Culture

HK-2 and L02 cells were cultured inRPMI-1640, while 4T1 and HepG2 cells were cultured in DMEM medium with 10% FBS (Gibco, Shanghai, China) and 1% penicillin/streptomycin (Gibco, Shanghai, China). All cell lines were purchased from Cell Bank of Shanghai Institute of Biochemistry and Cell Biology, Chinese Academy of Sciences (Shanghai, China), which was cultured in the incubator containing 5% CO_2_ at 37 °C.

### 5.3. Morphological Analysis of Cells

The density of 1 × 10^5^ cells/mL 4T1 cells (1 mL/well) were seeded in 6-well plates and incubated overnight in the atmosphere of 5% CO_2_ at 37 °C, followed by exposure to different concentrations of resveratrol (50, 100, 150, 200, and 250 µM) for 24, 48, and 72 h, respectively. Cell morphology was observed under an inverted microscope (Olympus Corporation, Beijing, China). For fluorescence excitation, a 460–495 nm excitation filter was used for green fluorescence and a 530–550 nm excitation filter for red fluorescence detection. 

### 5.4. Cell Viability Measurement

A rapid and highly sensitive assay based in the chromogenic reaction of WST-8 (a tetrazolium salt), Cell Counting Kit-8 (CCK-8) assay (Bioworld Technology, Nanjing, China), was used to evaluate cell proliferation and cytotoxicity. Cells at a density of 5 × 10^4^ cells/mL (100 µL/well) were seeded in 96-well plates and incubated overnight under 5% CO2 at 37 °C, followed by exposure to a series of concentrations of resveratrol (50, 100, 150, 200, and 250 µM). At the same time, a group only containing culture medium was set as blank control. Each group had six biological repeats. After dosing for 24, 48, and 72 h, the cells were washed and then fresh medium (100 µL) supplemented with 10 µL CCK-8 solution (0.5 mg/mL) was added to each well. After incubated in the dark for 2 h at 37 °C, the optical density at 450 (OD450) of each well were measured by plate reader (Synergy H4: Bio-Tek, Winooski, VT, USA). The results are presented as mean ± standard deviation (SD). The survival rate of control cells treated with 0 M resveratrol was set as 100%. Cell viability was calculated using the following Equation (1): (1)$$Cell\;viability\left(\%\right)=\frac{dosing\;cell\;OD-blank\;OD}{control\;cell\;OD-blank\;OD}\times 100$$

### 5.5. AO/EB Staining

Morphological apoptosis of 4T1 cells treated with different resveratrol concentrations for different times was assessed using an acridine orange/ethidium bromide (AO/EB) staining kit (Solarbio, Beijing, China). The density of 1 × 10^5^ 4T1 cells/mL were plated in 6-well plates (1 mL/well) and incubated overnight. The medium was replaced with RSV-containing (50, 100, 150, 200, and 250 µM) medium and incubated for 48 or 72 h under the same conditions mentioned before. Cells were washed with PBS and stained with AO/EB solution (20 µL AO/EB freshly mixed solution of equal volume in 1 mL PBS) for 2–3 min in the dark. After the successive washes, the fluorescent images were taken with inverted fluorescence microscope (Olympus Corporation, Beijing, China).

### 5.6. Cell Apoptosis and Cell Cycle Detection

The Annexin-V/PI Apoptosis Analysis Kit (Yeasen, Inc., Shanghai, China) was used to detect cell apoptosis. 4T1 cells were pretreated with different concentrations of resveratrol (50, 100, 150, 200, and 250 µM) for 48 h, or treated with IC50 for 0, 12, 24, 36, and 48 h, respectively. Cells collected by centrifugation were washed in precooled PBS and stained with Annexin V/Alexa Fluor 647 and propidium iodide (PI) according to the manufacturer’s protocol. The Cell Cycle Analysis Kit (Yeasen, Inc., Shanghai, China) was used to detect cell cycle progression. Cells were pretreated with IC50 of resveratrol for different times (0, 12, 24, 36, and 48 h). Cells collected by centrifugation were washed in precooled PBS and stained with propidium iodide (PI) according to the manufacturer’s protocol. Each sample was repeated three times. Data were acquired by the flow Cytometer (Accuri C6 Plus; BD Pharmingen, Shanghai, China) and analyzed by FlowJo-V10 software (Tree Star Inc, Ashland, OR, USA).

### 5.7. RNA-Sequencing and Transcriptome Analysis

The 4T1 cells treated with 0 M and IC50 (93 µM) for 24 h was set as control group (Mock) and treatment group (RSV_treated), respectively. Three replicates for each group were set up and total RNA extracted and sent to Shanghai Majorbio Bio-Pharm Technology Co., Ltd for transcriptome sequencing by Illumina HiSeqTM 2500 sequencer. The data were analyzed by the free online platform of Majorbio Cloud Platform [57].

### 5.8. qRT-PCR Verification

Total RNA was extracted from cells with TRIzol (Invitrogen, Carlsbad, CA, USA) according to the manufacturer’s protocols. Reverse transcription was performed using HiScript Q RT Super Mix for qPCR (Vazyme, Nanjing, China), and 500 ng of cDNA was used for real time qPCR with SYBR Green Master Mix (Vazyme) and the primers (Table A1). The expression levels of target genes were analyzed with Ppia as an internal control. Fold change was calculated using the 2^−ΔΔCt^ method. The results are presented as mean ± standard deviation (SD), and all samples were measured independently in triplicate.

### 5.9. Statistical Analysis

All data are presented as the mean ± standard error. A Student’s *t*-test or two-way ANOVA with Bonferroni correction (data and statistical analysis have *n* of at least three per group) were performed to determine statistical significance among groups in each of three independent experiments using GraphPad Prism 7 (GraphPad Software, Inc., San Diego, CA, USA, Version 7.00, 2017). *p* values < 0.05 were considered to indicate statistical significance [58]. Statistical significance of results is indicated in each figure.

### 5.10. Image Analysis

An Olympus IX73 microscope equipped with a 130W U-HGLGPS mercury lamp was used to obtain fluorescence images, which were recorded with an Olympus DP80 camera system and processed with the Olympus Cell Sens software. Images were collected in the 1360 × 1024 pixel format and cropped by Adobe Illustrator CC software to show the field of cells representative of the effect of treatment. Image J software (National Institutes of Health (NIH), Bethesda, MD, USA; Version 1.48, 2014) was used for cell counting and cell size measurement.

## Figures and Tables

**Figure 1 toxins-11-00731-f001:**
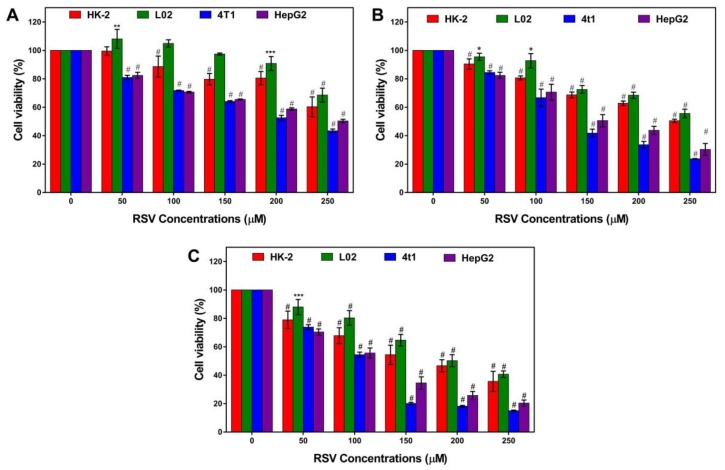
Effects of resveratrol on cell viability. (**A**) 24 h, (**B**) 48 h, and (**C**) 72 h cell viability of HK-2 (red), L02 (green), 4T1 (blue), HepG2 (magenta) treated with a serial concentration of resveratrol. Error bars are standard deviations. Significant differences are indicated as **p* < 0.05, ** *p* < 0.01, *** *p* < 0.001, and ^#^
*p* < 0.0001 (or **** *p* < 0.0001).

**Figure 2 toxins-11-00731-f002:**
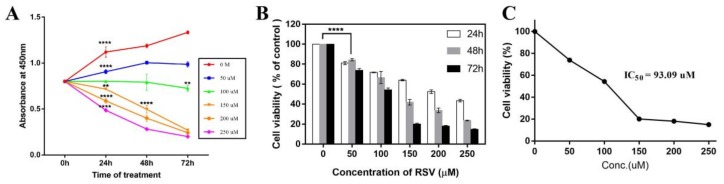
Effects of resveratrol on proliferation and viability of 4T1 cells. (**A**) Time-dependent proliferation curve of 4T1 cells after treating with different concentrations of resveratrol (RSV) as measured by CCK-8 assay. (**B**) Cell viability of 4T1 after treating with different concentrations of RSV for different time periods. (**C**) Dose-dependent viability curve at 72 h. Error bars are standard deviations. Significant differences are indicated as * *p* < 0.05, ** *p* < 0.01, *** *p* < 0.001, and **** *p* < 0.0001.

**Figure 3 toxins-11-00731-f003:**
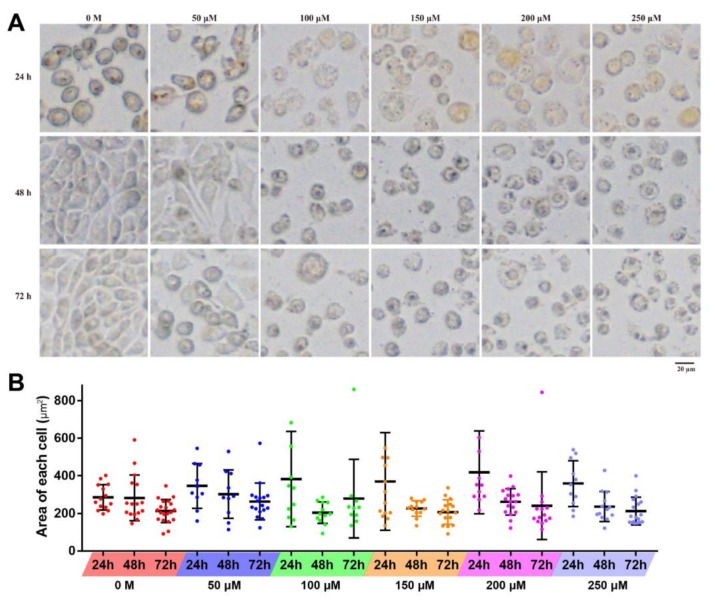
Effects of resveratrol on the morphology of 4T1 cells. (**A**) 4T1 cells were treated with different concentrations of resveratrol (50, 100, 150, 200, and 250 µM) for 24, 48, and 72 h, respectively, then morphological changes were observed. Magnification: 100×. (**B**) The statistical diagram of the area of 4T1 cells. Error bars are standard deviations.

**Figure 4 toxins-11-00731-f004:**
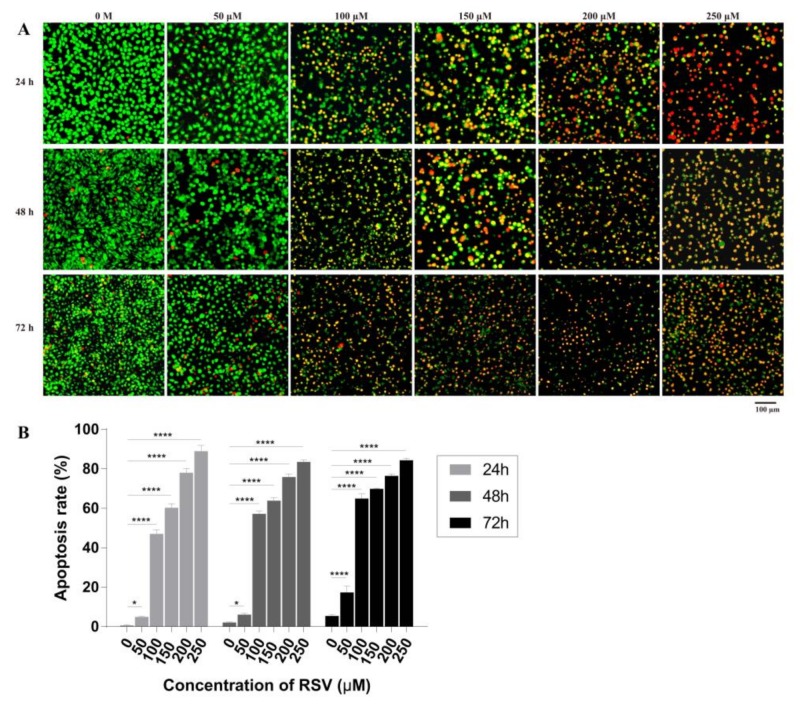
Effects of resveratrol treatment on apoptosis of 4T1 cells. (**A**) Apoptosis of 4T1 cells induced by different concentrations of resveratrol at different times (24, 48, and 72 h) detected by acridine orange (AO)/ethidium bromide (EB) staining by fluorescence microscope. Magnification: 100×. (**B**) Apoptotic rate of 4T1 cells calculated according to the results of AO/EB staining. The results are presented as mean ± standard deviation (SD), and all samples were measured independently in triplicate. Error bars are standard deviations. Significant differences are indicated as * *p* < 0.05 and **** *p* < 0.0001.

**Figure 5 toxins-11-00731-f005:**
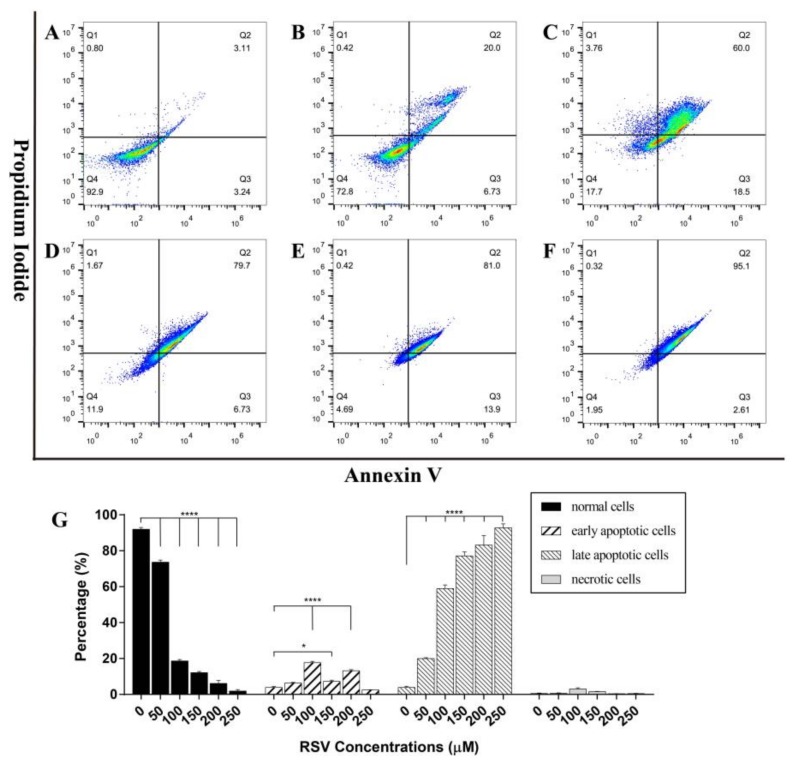
The apoptosis of 4T1 cells was detected by flow cytometry. (**A**–**F**) Apoptosis stage diagram of 4T1 cells after 48 h treatment with different concentrations of resveratrol (A: 0 M; B: 50 µM; C: 100 µM; D: 150 µM; E: 200 µM; F: 250 µM). Cells in Q1, Q2, Q3, and Q4 quadrants are necrotic cells, late apoptotic cells, early apoptotic cells, and normal cells, respectively. (**G**) The ratio histogram of different cell types. The results are presented as mean ± standard deviation, and all samples were measured independently in triplicate. Error bars are standard deviations. Significant differences are indicated as * *p* < 0.05 and **** *p* < 0.0001.

**Figure 6 toxins-11-00731-f006:**
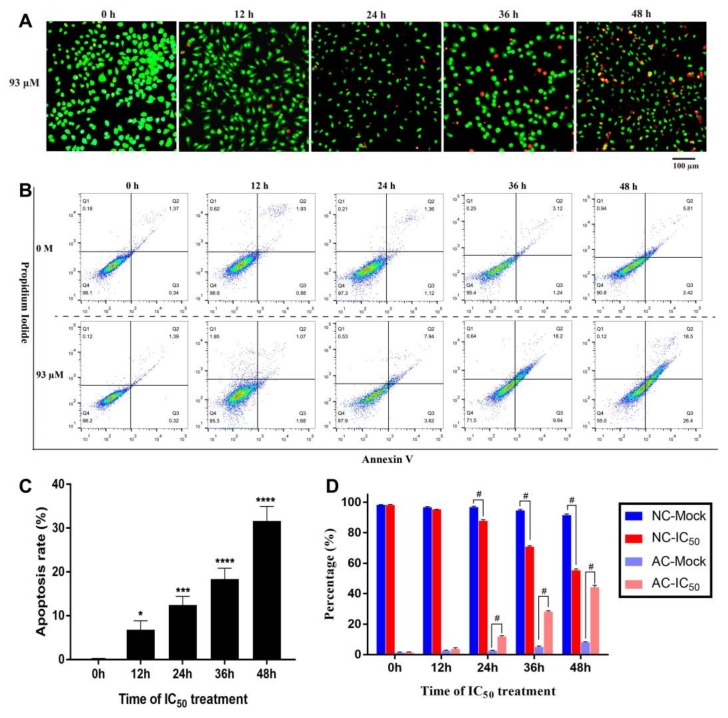
Apoptosis of 4T1 cells treated with IC50 of resveratrol. (**A**) After treating 4T1 cells with IC50 (93 µM) for different time (0, 12, 24, 36, and 48 h), AO/EB staining solution was used to detect cell morphological apoptosis. Magnification: 100×. (**B**) The apoptotic stages of 4T1 cells were detected and analyzed by flow cytometry under the treatment of IC50 (93 µM) or control (0 µM) for different time. Cells in Q1, Q2, Q3, and Q4 quadrants are necrotic cells, late apoptotic cells, early apoptotic cells, and normal cells, respectively. (**C**) The morphological apoptotic rate of 4T1 cells was calculated according to the results of AO/EB staining. (**D**) The proportion of normal cells and apoptotic cells was calculated according to the results of apoptotic stage detection. In this picture, NC stands for normal cells, AC stands for apoptotic cells, Mock stands for control group, and IC50 stands for the 93 µM resveratrol treatment group. The results are presented as mean ± standard deviation, and all samples were measured independently in triplicate. Error bars are standard deviations. Significant differences are indicated as * *p* < 0.05, ** *p* < 0.01, *** *p* < 0.001 and ^#^*p* < 0.0001 (or **** *p* < 0.0001).

**Figure 7 toxins-11-00731-f007:**
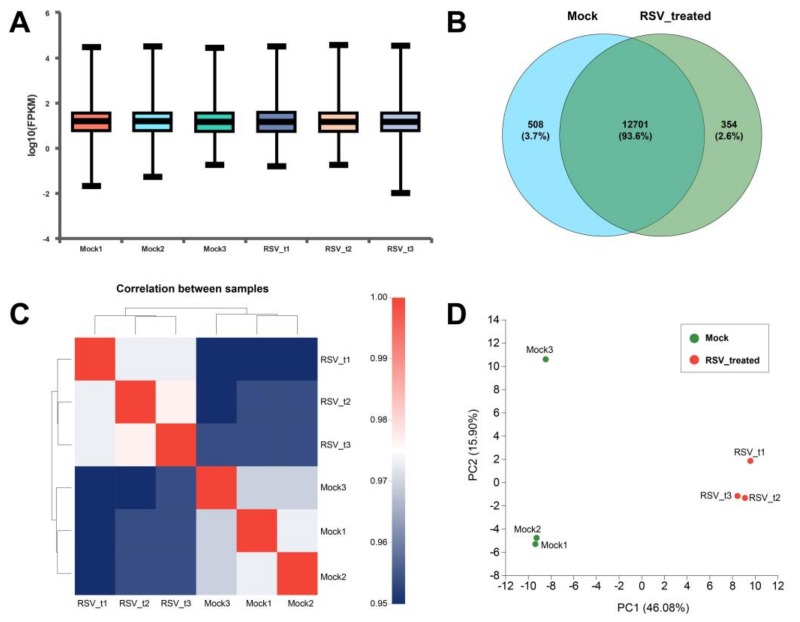
Expression analysis and relationship analysis of six samples. (**A**) Expression statistics of six samples. The horizontal line in each sample shows the median expression in the sample. (**B**) Venn analysis between Mock and RSV_treated groups. (**C**) Correlation analysis among six samples. The squares of different colors represent the degree of correlation between the two groups. (**D**) Principal component analysis (PCA) between Mock and RSV_treated groups. The closer each sample point is, the higher the similarity is.

**Figure 8 toxins-11-00731-f008:**
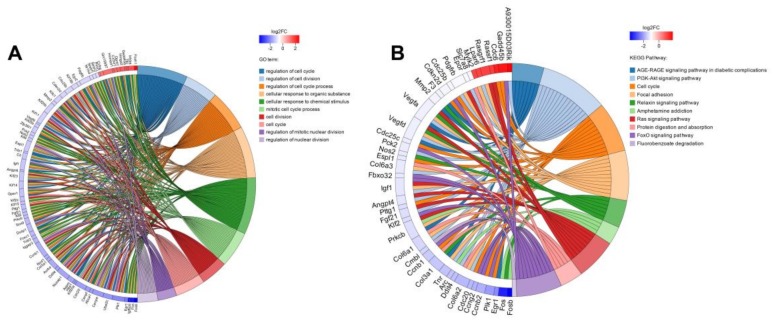
Enrichment string diagrams of differentially expressed genes (DEGs). (**A**) GO enrichment string diagram. (**B**) KEGG enrichment string diagram. The left side of the ring structure is the gene name, which is arranged in the order of log_2_FC. The larger log_2_FC is, the larger the gene expression difference factor is. On the right of the ring is the significantly enriched GO Term or KEGG pathway information for differentially expressed genes.

**Figure 9 toxins-11-00731-f009:**
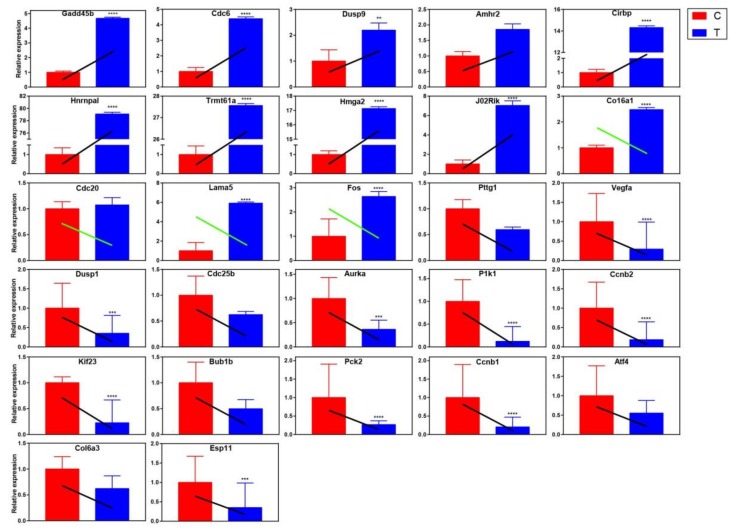
The expression of the selected DEGs relative to internal reference gene was quantified by qRT-PCR. Group C is the control group, and group T is the resveratrol treatment group. The colored rectangle in the figure is the relative expression of each gene calculated by results of qRT-PCR, while the oblique line represents the relative expression trend of each gene obtained by transcriptional sequencing. The black slash indicates that the sequencing results are consistent with the qRT-PCR validation results, while the green slash indicates that the sequencing results are contrary to the qRT-PCR validation results. All qRT-PCR data were shown as mean ± standard deviation; *n* = 3. Significant differences are indicated as * *p* < 0.05, ** *p* < 0.01, *** *p* < 0.001, and **** *p* < 0.0001.

**Figure 10 toxins-11-00731-f010:**
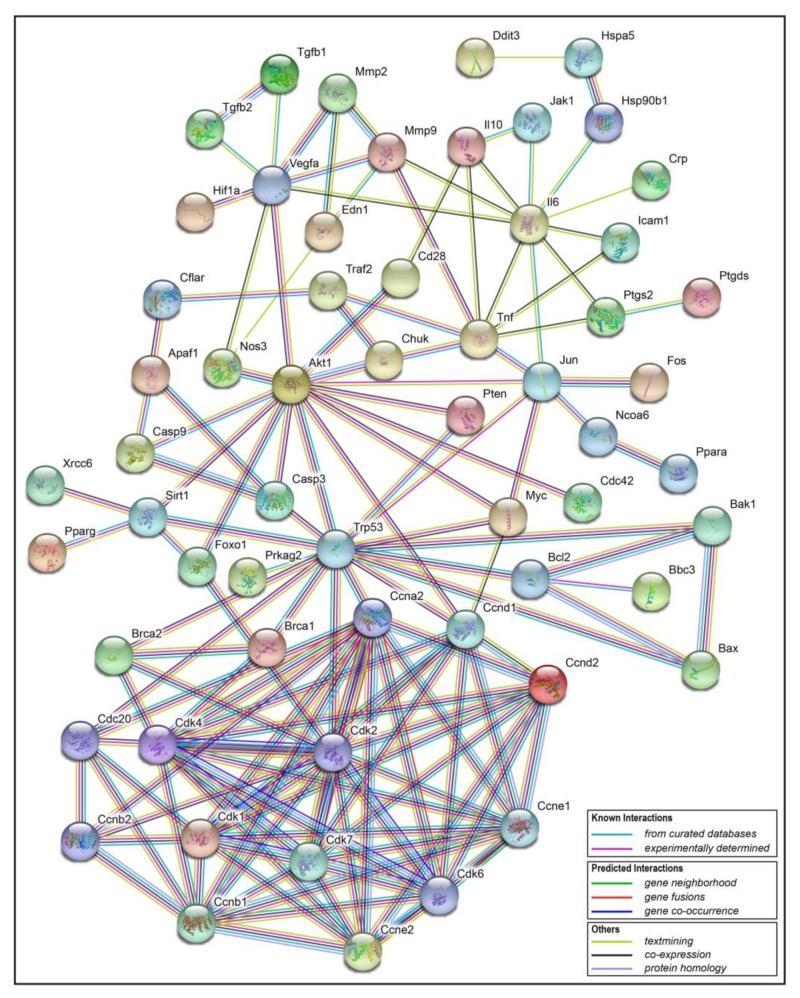
Target protein-protein interaction (PPI) network. In this network, nodes represent proteins, empty nodes represent proteins of unknown 3D structure, filled nodes represent some 3D structure is known or predicted; edges represent protein-protein associations. The specific interactions are explained in the bottom rectangle.

**Figure 11 toxins-11-00731-f011:**
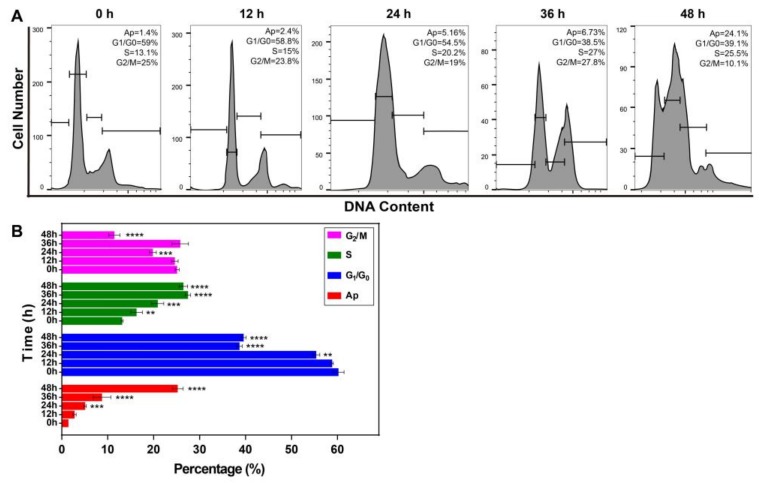
Effects of resveratrol on cell cycle alteration. (**A**) 4T1 cells were treated with IC50 (50% inhibiting concentration) for different times (0, 12, 24, 36, and 48 h), and cell cycle phases were analyzed by flow cytometry. Ap, Apoptosis phase; G1/G0, DNA presynthetic phase and stationary phase; S, DNA synthesis phase; G2/M, DNA postsynthetic phase and mitotic phase. (**B**) The percentage of cells at different time points was calculated. Error bars are standard deviations. Significant differences are indicated as * *p* < 0.05, ** *p* < 0.01, *** *p* < 0.001, and **** *p* < 0.0001.

**Figure 12 toxins-11-00731-f012:**
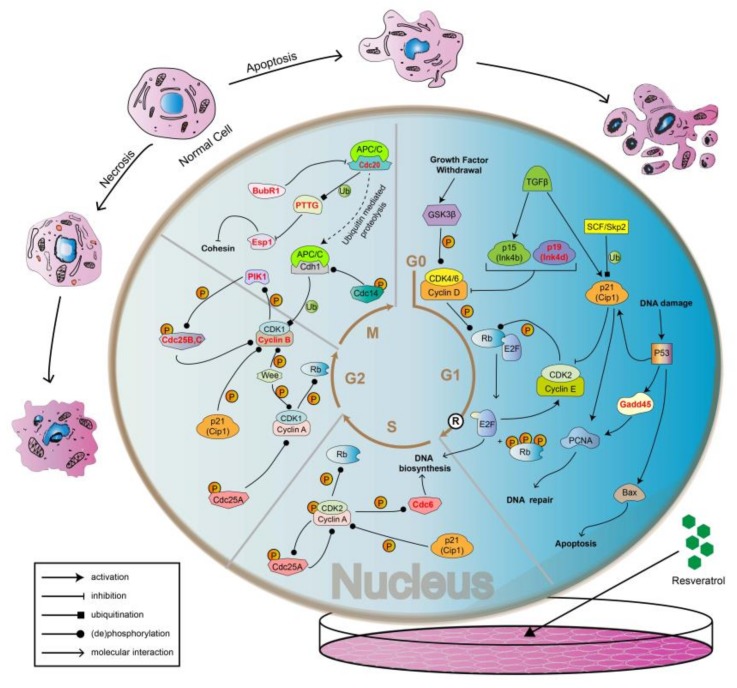
The regulatory mechanism of cell cycle progression. The box at the bottom left is a legend of the action mode. In each phase of the cell cycle (G1, S, G2, and M phase), the red bolded molecules are the differentially expressed genes detected by the transcriptome in our study.

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
