# Peer review of "The Cytotoxicity Effect of Resveratrol: Cell Cycle Arrest and Induced Apoptosis of Breast Cancer 4T1 Cells"

_toxins, 2019, doi:10.3390/toxins11120731_

Round 1

Reviewer 1 Report

The journal covers “toxinology and all kinds of toxins (biotoxins) from animals, microbes and plants”. Although the definition of Toxin is not consensual among toxicologists, I consider that the topic of the submitted paper is outside the scope of the journal. Therefore, although the paper is interesting and has scientific quality, I recommend authors to submit it to another journal. The areas of cancer or nutrition would fit.

Regardless the scope, the manuscript is generally well written. However, some aspects should be modified/clarified in the paper:

Line 58 – replace cell activity by cell metabolic activity

The number of independent experiments is missing in many cases. Please clarify how many replicates and how many independent experiments were performed.

lines 67-68 – specify the type of each cell line (ex., hepatocyte, kidney cell)

line 83 – negative growth is not an accurate term

Figure 2A – It is not clear where these data come from. How was the experiment done?

Fig. 4 B, Fig. 5 – please check statistics. It is unlikely that there is significance between 0 and 50/100, but not for higher concentrations

Lines 135-139 and 299-305 – remove; too much detail for a widely known technique

Section 2.6 – sub-G1 population does not necessarily means apoptosis; please revise text. The S population can only be adequately quantified using a DNA replication marker

Line 328 – replace the word “believed”

lines 412-419 - remove; too much detail; standard procedure of trypsinization

line 426 - mention that it is a tetrazolium salt-based assay

Author Response

Point 1:  English language and style are fine/minor spell check required 

Response 1: English language and style have been thoroughly edited throughout this manuscript.

Point 2:  The topic of the submitted paper is outside the scope of the journal. The areas of cancer or nutrition would fit.

Response 2: Thank you for your kind advice, we agree with this comment that our manuscript would be better in the journal with the topic of cancer. However, some recent papers in this journal have shown that our article is also suitable for Toxins. As examples see:

Chen et al. Momordica charantiaRibosome-Inactivating Protein α-Momorcharin Derived from Edible Plant Induces Inflammatory Responses by Activating the NF-kappaB and JNK Pathways.[J] .Toxins (Basel), 2019, 11: undefined. Ruiz-de-la-Herrán Javier et al. Inclusion of a Furin Cleavage Site Enhances Antitumor Efficacy against Colorectal Cancer Cells of Ribotoxin α-Sarcin- or RNase T1-Based Immunotoxins.[J] .Toxins (Basel), 2019, 11: undefined. Fuchs Hendrik, Dianthin and Its Potential in Targeted Tumor Therapies.[J] .Toxins (Basel), 2019, 11: undefined. Chichirau Bianca E et al. Helicobacter pyloriTyrosine Kinases in Infections and Gastric Cancer.[J] .Toxins (Basel), 2019, 11: undefined.

Point 3: Line 58 – replace cell activity by cell metabolic activity

Response 3: As suggested by the reviewer, we have replaced cell activity by more accurate description“cell metabolic activity”as followings (pg.2, line 58):

This study calculated IC50 (50% inhibiting concentration) of resveratrol in 4T1 breast cancer cell lines by detecting cell metabolic activity.

Point 4: The number of independent experiments is missing in many cases. Please clarify how many replicates and how many independent experiments were performed.

Response 4: We are very sorry for our negligence at this point,  and we have therefore added  "statistical analysis" part in the section of 5. Materials and Methods to specify the replications and the number of independent experiments. As followings (pg.16, line 480-485):

5.9. Statistical analysis.

All data are presented as the mean±standard error. A Student's t-test or two-way ANOVA with Bonferroni correction (data and statistical analysis have n of at least 3/group) were performed to determine statistical significance among groups in each of three independent experiments using GraphPad Prism 7 (GraphPad Software). p values < 0.05 were considered to indicate statistical significance (Grimaldi Maddalena, 2019). Statistical significance of results was indicated in each figure.

Point 5: lines 67-68 – specify the type of each cell line (ex., hepatocyte, kidney cell)

Response 5: According to the review’s suggestion, the type of each cell line has been  described in detail as followings (pg.2, line 67-69 ):

We have evaluated the cytotoxic effect of resveratrol on two types of normal cell lines (the renal tubular epithelial cell line HK-2 and normal human liver cell line L02), and two types of tumor cell lines (hepatocellular carcinoma HepG2 and murine mammary carcinoma cell line 4T1) (Fig.1).

Point 6: line 83 – negative growth is not an accurate term

Response 6: We appreciate the review’s attention to detail, and we have corrected the description as suggested. As followings (pg.3, line 84):

When the treatment concentration was higher than 100 µM, 4T1 cells showed inhibitory growth.

Point 7: Figure 2A – It is not clear where these data come from. How was the experiment done?

Response 7: Sorry for confusion, and we have modified the ambiguous text to be more clear.

The revision in Figure 2A (pg.3, line 90-92) as followings:

Time-dependent proliferation curve of 4T1 cells after treating with different concentrations of resveratrol (RSV) as measured by cck-8 assay.

The specific experimental procedure was described in section  5.4. Cell Viability measurement  (pg.15, line 432-444).

Point 8: Fig. 4 B, Fig. 5 – please check statistics. It is unlikely that there is significance between 0 and 50/100, but not for higher concentrations.

Response 8: We thank the reviewer for details. We have checked the statistics of the significant differences among the six groups (0, 50, 100, 150, 200, and 250 µM) at different time points (24 h, 48 h, and 72 h), and found that there is no problem with our data and analysis. We chose two way ANOVA (Sidak's multiple comparisons test and Dunnett's multiple comparisons test) to analyze those data (apoptotic rate of AO/EB staining (Fig.4B) and the ratio of cell types detected by flow cytometry (Fig.5)). Our analysis shows that the p-value is less than 0.05 between 0 and 50/100, indicating statistical significance. The details about the analysis of Fig.4B (Sidak's multiple comparisons test) and Fig.5B (Dunnett's multiple comparisons test) are provided in supplementary materials.

Point 9: Lines 135-139 and 299-305 – remove; too much detail for a widely known technique

Response 9: We have removed the previous text “Among them, double negative cells that could ..., which located in Q1 quadrant.” and “The DNA content in G1/G0 phase is... in the cell cycle phase distribution diagram.” as suggested.

Point 10: Section 2.6 – sub-G1 population does not necessarily means apoptosis; please revise text.

Response 10: We are very sorry for our incorrect writing. We agree with your point of view that sub-G1 peak is not a sufficient and necessary condition for apoptosis. The occurrence of sub-G1 peak is due to the break and shrinkage of the DNA fragment, resulting that the fluorescence intensity of PI staining is smaller than that of G1 phase, which belongs to the normal diploid phase, so called it sub-diploid peak. If the broken DNA could not get out of the cell membrane, sub-G1 will be difficult to appear. Therefore, the sub-G1 peak of PI staining is not as valuable as the apoptosis detection of Annexin V/PI. PI itself is mainly used to detect the changes in cell cycle, not apoptosis. According to suggestion of Point 9, we have removed the content about sub-G1 and we promise to take note of this point in the future writing. (pg11, line 296-308)

Point 11: The S population can only be adequately quantified using a DNA replication marker.

Response 11: The reviewer has made a very good point here. The quantification of the S phase does depend on DNA replication markers. In this paper, we just tested the changes of cell cycle in the current study. If we need to accurately quantify the S-phase DNA in the future work, using replication markers would be a good choice, which would provide valuable information to the study. So we discussed about it in the “3. Discussion” section and wish to finish it in our continued study. As followings (pg.13, Line 345-353):

In terms of monitoring the cell cycle, in addition to the flow cytometry analysis of PI staining used in this study, immunostaining evaluation of cell cycle markers is also an effective method, among which MCM2, ki-67 and PCNA are common used (GuziÅ„ska-Ustymowicz Katarzyna, 2009). Compared with PCNA and ki-67, the advantage of MCM is that it is expressed throughout the cell cycle and it is immune to DNA damage repair and other external factors (Wojnar A, 2011). Therefore, MCM protein is considered to be an ideal marker of cell proliferation superior to PCNA and ki-67. However, from the practical application level, PCNA and ki-67 are more widely used. Unfortunately, the methods used to detect cell proliferation, apoptosis , and cell cycle in this paper are relatively simple. In future experiments, different methods mentioned above should be tried to reach more profound conclusions.

Point 12: Line 328 – replace the word “believed”

Response 12: We appreciate the review’s attention to detail, and we have replaced the word “believed” by the word “found”.

Point 13: lines 412-419 - remove; too much detail; standard procedure of trypsinization

Response 13: We have removed the previous text about standard procedure of trypsinization as suggested.

Point 14: line 426 - mention that it is a tetrazolium salt-based assay

Response 14: Thank you for your suggestion. We have modified the text and mentioned the tetrazolium salt-based assay to make it to be more clear. As followings (pg.15, Line 433-435):

The Cell Counting Kit-8 (CCK-8) assay (Bioworld Technology, Nanjing, China), which is a rapid and highly sensitive assay kit based in the chromogenic reaction of WST-8 (a tetrazolium salt) to evaluate cell proliferation and cytotoxicity.

Special thanks to you for your good comments. 

Reviewer 2 Report

The cytotoxicity effect of resveratrol: cell cycle arrest and induced apoptosis of breast cancer 4T1 cells (Manuscript ID: toxins-659319).

In this paper, it has compared the activity of normal cell lines and cancer cell lines after treating with resveratrol. The authors found that, this natural polyterpenoid, present more significant cytotoxicity in 4T1 cell line. Moreover, they conclude that resveratrol could induce apoptotic cell death in a dose- and time-dependent manner, and this drug arrested the 4T1 cells in the S phase of the cell cycle.

After reading this manuscript, my view is that it is interesting and is well written. Data are original. The topic is timely. The study has also a basic importance in the clinical and experimental field of therapeutic strategies for the treatment of the breast cancer. The manuscript may be suitable for publication in toxins after major revision. There are several points that need attention:

1.-   A new subsection entitled "statistical analysis" should be provided. This could be included in the section 5. Materials and Methods. As example see:

Grimaldi M, Dal Bo V, Ferrari B, Roda E, De Luca F, Veneroni P, Barni S, Verri M, De Pascali SA, Fanizzi FP, Bernocchi G, Bottone MG (2019) Long-term after treatment with platinum compounds, cisplatin and [Pt(O,O'-acac )(γ-acac)(DMS)]: autophagy activation in rat B50 neuroblastoma cells. Toxicol Appl Pharmacol 364: 1-11.

Moreover, in figure 2, the authors should indicate wether data are presented as error bars and standard deviations or mean ± standard deviation. In addition to that, p values should be supplied.

2.- Page 3. Section entitled: Effects of resveratrol on the cell morphology (lines 93 to 105).

After observing figure 3, 4T1 cells seem to have different size. Are they smaller? In my view, the area of the 4T1 cells should be determined. The Image J software can be used.

3.- Page 4. Section entitled: Detection of cell apoptosis (lines 107 to 130).

The authors indicated: "thus live cells will appear uniformly green while early apoptotic cells will show condensed or fragmented nucleus with bright green color. Late apoptotic cells will show condensed and fragmented orange chromatin". In my view, this analyses is insufficient, 4T1 cells should be immunolabeled with selective antibodies to detect activated caspase-3 and caspase-9. As example, see:

Grimaldi M, Santin G, Insolia V, Dal Bo V, Piccolini VM, Veneroni P, Barni S, Verri M, De Pascali SA, Fanizzi FP, Bernocchi G, Bottone MG (2016) [Pt(O,O'-acac )(γ-acac)(DMS)] versus cisplatin: apoptotic effects in B50 neuroblastoma cells. Histochem Cell Biol 145: 587-601.

4.- Page 11. Section entitled: Resveratrol changed the phase distribution of the cell cycle (lines 296 to 319).

The authors assert that resveratrol can promote the transformation from G1 to S phase in breast cancer cells, and then induce cycle arrest in S phase, so that their proliferation capacity weakened and cell viability reduced. These results are very interesting but I advise the authors to study at least two markers: (I) the thymidine analog 5-bromo-2'deoxyuridine (an indicator of DNA synthesis) and (II) the proliferating cell nuclear antigen (PCNA), which begins to accumulate during the G1 phase of the cell cycle, is most abundant during the S phase, and declines during the G2/M phase.

5.- Page 15. Section entitled: Morphological analysis of cells (lines 420-424).

The authors indicated: "cell morphology was observed under an inverted microscope (Olympus Corporation, Japan)". They should indicate characteristics of fluorescence excitation.

6.-   A new subsection entitled "image analysis" should be provided. This could be included in the section 5. Materials and Methods. This subsection should provide several characteristics of the fluorescence microscopy such as the name of the digital camera used for recorded, the order of magnitude of the optic analysis, .......

Author Response

Response to Reviewer #2

We want to thank Reviewer #2 for constructive and insightful criticism and advice. We addressed all the points raised by the reviewer as summarized below.

Point 1:  English language and style are fine/minor spell check required 

Response 1: English language and style have been thoroughly edited throughout this manuscript.

Point 2: A new subsection entitled "statistical analysis" should be provided. This could be included in the section 5. Materials and Methods.

Response 2: Considering the Reviewer’s suggestion, we have added  "statistical analysis" subsection in the section 5. Materials and Methods. As followings (pg.16, line 480-485):

5.9. Statistical analysis.

All data are presented as the mean±standard error. A Student's t-test or two-way ANOVA with Bonferroni correction (data and statistical analysis have n of at least 3/group) were performed to determine statistical significance among groups in each of three independent experiments using GraphPad Prism 7 (GraphPad Software). p values < 0.05 were considered to indicate statistical significance (Grimaldi Maddalena, 2019). Statistical significance of results was indicated in each figure.

Point 3: In figure 2, the authors should indicate wether data are presented as error bars and standard deviations or mean ± standard deviation. In addition to that, p values should be supplied.

Response 3: Thank the reviewer for this insightful comment. We have now analyzed the significant differences among groups. The information of p values and error bars has been added into the figure and is also supplied in the legend of Figure 2 (pg.3, line 89-95).

Point 4: Page 3. Section entitled: Effects of resveratrol on the cell morphology (lines 93 to 105).

After observing figure 3, 4T1 cells seem to have different size. Are they smaller? In my view, the area of the 4T1 cells should be determined. The Image J software can be used.

Response 4: In response to the reviewer’s requests, we have studied how to use  Image J software to measure the area of the 4T1 cells. We found that the area of the 4T1 cells is becoming smaller as supposed. The new data are depicted in Fig. 3B “ The statistical diagram of the area of 4T1 cells” (pg.4). The corresponding description was added as followings  (pg.3, line 104-105):

Furthermore, with the increase of treatment time, the area of most observed cells gradually decreased at each concentration, except for those swelling and rupture cells (Fig.3B).

Point 5: Page 4. Section entitled: Detection of cell apoptosis (lines 107 to 130). The authors indicated: "thus live cells will appear uniformly green while early apoptotic cells will show condensed or fragmented nucleus with bright green color. Late apoptotic cells will show condensed and fragmented orange chromatin". In my view, this analyses is insufficient, 4T1 cells should be immunolabeled with selective antibodies to detect activated caspase-3 and caspase-9.

Response 5: This is a valid suggestion, however, we also used other methods to demonstrate the occurrence of apoptosis in this paper, such as flow cytometry and transcriptome sequencing. Of course, there is no doubt that the detection of  activated caspase-3 and caspase-9 has been proved to be a better method in apoptosis detection, and we will be actively to take advantage of this method for further study in our lab. Therefore, we discussed about it in the “3.Discussion” section and wish to finish it in our continued study. As followings (pg.12-13, line 342-345):

In the detection of apoptosis at the molecular level, the normally detected proteins are caspase-9, caspase-3, and PARP (substrates of caspase-3), located in the mitochondrial pathway (Grimaldi Maddalena, 2016). In addition, the ratio of Bcl-2 /Bax could also be used to represent the degree of apoptosis (He Yingying, 2019).

Point 6: Page 11. Section entitled: Resveratrol changed the phase distribution of the cell cycle (lines 296 to 319). The authors assert that resveratrol can promote the transformation from G1 to S phase in breast cancer cells, and then induce cycle arrest in S phase, so that their proliferation capacity weakened and cell viability reduced. These results are very interesting but I advise the authors to study at least two markers: (I) the thymidine analog 5-bromo-2'deoxyuridine (an indicator of DNA synthesis) and (II) the proliferating cell nuclear antigen (PCNA), which begins to accumulate during the G1 phase of the cell cycle, is most abundant during the S phase, and declines during the G2/M phase.

Response 6: These two markers recommended by the reviewer are very interesting and insightful. We believe that the detection of them would provide additional arguments to our paper, so we decided to discuss these two markers in the “3.Discussion” section and apply them in the future work. As followings (pg.12-13, line 337-353):

When detecting cell proliferation at the cellular level, CCK8 staining and Brdu (the thymidine analogues) staining are usually used. However, these end-point assay need to destroy cells, leading to cell death or the destruction of cell structure. The real time cell analyzer (RTCA, xCELLigence, Roche) is an impedance-based technology that can be used for label-free and real-time monitoring of cellproperties, such as cell adherence, proliferation, migration and cytotoxicity, which could be widespread used (Roshan Moniri Mani, 2015). In the detection of apoptosis at the molecular level, the normally detected proteins are caspase-9, caspase-3, and PARP (substrates of caspase-3), located in the mitochondrial pathway (Grimaldi Maddalena, 2016). In addition, the ratio of Bcl-2 /Bax could also be used to represent the degree of apoptosis (He Yingying, 2019). In terms of monitoring the cell cycle, in addition to the flow cytometry analysis of PI staining used in this study, immunostaining evaluation of cell cycle markers is also an effective method, among which MCM2, ki-67 and PCNA are common used (GuziÅ„ska-Ustymowicz Katarzyna, 2009). Compared with PCNA and ki-67, the advantage of MCM is that it is expressed throughout the cell cycle and it is immune to DNA damage repair and other external factors (Wojnar A, 2011). Therefore, MCM protein is considered to be an ideal marker of cell proliferation superior to PCNA and ki-67. However, from the practical application level, PCNA and ki-67 are more widely used. Unfortunately, the methods used to detect cell proliferation, apoptosis , and cell cycle in this paper are relatively simple. In future experiments, different methods mentioned above should be tried to reach more profound conclusions.

Point 7: Page 15. Section entitled: Morphological analysis of cells (lines 420-424). The authors indicated: "cell morphology was observed under an inverted microscope (Olympus Corporation, Japan)". They should indicate characteristics of fluorescence excitation.

Response 7: We are very sorry for our negligence of the fluorescence excitation. According to the suggestion, we have now provided the parameters of our fluorescence excitation. As followings (pg.15, line 429-431):

For fluorescence excitation, a 460-495 nm excitation filter was used for green fluorescence and a 530-550 nm excitation filter for red fluorescence detection.

Point 8: A new subsection entitled "image analysis" should be provided. This could be included in the section 5. Materials and Methods. This subsection should provide several characteristics of the fluorescence microscopy such as the name of the digital camera used for recorded, the order of magnitude of the optic analysis.   

Response 8: As reviewer suggested that we have added "image analysis" subsection to offer several characteristics of the fluorescence microscopy at the end of the section 5. Materials and Methods. As followings (pg.16, line 486-491):

5.10. Image analysis

An Olympus IX73 microscope equipped with a 130W U-HGLGPS mercury lamp was used to obtain fluorescence images, which were recorded with an Olympus DP80 camera system and processed with the Olympus Cell Sens software. Images were collected in the 1360×1024 pixel format and cropped by Adobe Illustrator CC software to show the field of cells representative of the effect of treatment. Image J software was used for cell counting and area measure.

Once again, thank you very much for your comments and suggestions.

Round 2

Reviewer 1 Report

Authors have duly addressed my comments. I only have one remaining point regarding the statistics of figures 4B and 5. As I mentioned previously, it is unlikely that there is significance between 0 and 50/100, but not for higher concentrations. Indeed, authors sent the full statistics in supplementary material and there are also significant differences between 0 and higher concentrations (150, 200, and 250 µM). For coherence, statistical significance should be shown in the graphics. 

Author Response

Thank you very much for your comments and suggestions.

Point 1:For coherence, statistical significance should be shown in the graphics. 

Response 1:We have provided the statistical significance in  figure 4B and figure 5 according to the reviewers suggestion. (pg. 5, line 129; pg. 6, line 152)

Reviewer 2 Report

Having carefully read the revised version of the manuscript entitled "The cytotoxicity effect of resveratrol: cell cycle arrest and induced apoptosis of breast cáncer 4T1 (659319)", in my view, this is suitable for publication in Toxins.

Author Response

Special thanks to you for your good comments.